# CHATQA 2: BRIDGING THE GAP TO PROPRIETARY LLMS IN LONG CONTEXT AND RAG CAPABILITIES

**Peng Xu**[*]    **Wei Ping**[*]    **Xianchao Wu**    **Chejian Xu**

**Zihan Liu**    **Mohammad Shoeybi**    **Bryan Catanzaro**

NVIDIA

[*]{pengx, wping}@nvidia.com

## ABSTRACT

In this work, we introduce ChatQA 2, an Llama 3.0-based model with a 128K context window, designed to bridge the gap between open-source LLMs and leading proprietary models (e.g., GPT-4-Turbo-2024-04-09) in long context understanding and retrieval-augmented generation (RAG) capabilities. These two capabilities are complementary to each other and essential for LLMs to process large volumes of information that cannot fit into a single prompt. We present a detailed continued training recipe to extend the context window of Llama3-70B-base from 8K to 128K tokens, along with a three-stage instruction tuning process to enhance the model's instruction-following, RAG performance, and long-context understanding capabilities. Our results demonstrate that the `Llama3-ChatQA-2-70B` model outperforms most existing state-of-the-art models, including GPT-4-Turbo-2024-04-09, Qwen2-72B-Instruct, and Llama3.1-70B-Instruct, on ultra-long tasks beyond 100K tokens, as well as on the RAG benchmark using only a 4K context window, showing the strong long context capability across varying sequence lengths. We further provide extensive comparisons between direct long-context and RAG solutions using the same state-of-the-art long-context LLMs. Interestingly, we find that the performance of strong long-context LLMs using RAG improves when retrieving a larger number of chunks. With a large set of top-$k$ chunks, RAG consistently outperforms direct long-context solution using the same state-of-the-art long-context models (e.g., `Llama3-ChatQA-2-70B` and Qwen2-72B-Instruct) on both 32K and 128K benchmarks. We open-source the model weights, training data, and the evaluation setup for the for the community: https://chatqa2-project.github.io/

## 1 INTRODUCTION

The open LLM community has made significant progress in advancing the capabilities of open-access large language models (LLMs), including Llama-3-70B-Instruct (Meta-AI, 2024), Qwen2-72B-Instruct (Alibaba-QWen, 2024), Nemotron-4-340B-Instruct (NVIDIA et al., 2024), and Mixtral-8x22B-Instruct-v0.1 (Mistral, 2024). However, performance gaps compared to frontier proprietary models, e.g., GPT-4-Turbo (OpenAI, 2023), still exist in many domains. Additionally, open-access models focused on key domains have been developed, such as DeepSeek-Coder-V2 (Zhu et al., 2024) for coding and math, ChatQA 1.5 (Liu et al., 2024) for conversational QA and retrieval-augmented generation (RAG), and InternVL 1.5 (Chen et al., 2024) for vision-language tasks, which can be on par with GPT-4-Turbo-2024-04-09 (OpenAI, 2023) in the certain domains.

In recent developments, the trend of extending the context window length in LLMs has gained remarkable traction within both the industrial and research communities. All leading proprietary LLMs support very large context window, allowing them to accommodate several hundred pages of text in a single prompt. For example, GPT-4 Turbo (OpenAI, 2023) and Claude 3.5 Sonnet offer a 128K and 200K context window, respectively. Meanwhile, Gemini 1.5 Pro (Gemini-Team, 2024) impressively supports up to a 10M context. Open-access LLMs have also made significant strides to keep up (01.AI et al., 2024; Alibaba-QWen, 2024). For instance, Qwen2-72B-Instruct (Alibaba-QWen, 2024) and Yi-34B (01.AI et al., 2024) support 128K and 200K context windows, respectively.

| Model Type | Long Context LLM (128K) | Ultra-long (>100K) 4 tasks | Long (32K) 6 tasks | Short (4K) 9 tasks (RAG) |
|---|---|---|---|---|
| Proprietary | GPT-4-Turbo-2024-04-09 | 33.16 | **51.93** | 54.72 |
| OPEN ACCESS | Qwen2-72B-Instruct | 39.77 | 49.94 | 54.06 |
| | Llama3.1-70B-Instruct | 39.81 | 49.92 | 52.12 |
| Open Source | Llama3-ChatQA-2-70B | **41.04** | 48.15 | **56.30** |

Table 1: `Llama3-ChatQA-2-70B` versus state-of-the-art long context LLMs across tasks with varying context lengths. Notably, *we open-source the training data and reproduction recipe* for building 128K long-context LLMs from pretrained base models with 8K context, resources that are not currently available for open-access (open-weights) 128K context models like Qwen2 and Llama3.1. Our model achieves the highest average score on four real-world ultra-long context (beyond 100K) tasks from InfiniteBench, and nine short context (within 4K) tasks from CHATRAG BENCH.

The concurrent work Llama-3.1-70B-Instruct (Meta-AI, 2024) also supports a 128K context window. However, the absence of training data and reproduction recipe makes replicating these models challenging. In addition, these models have mostly been evaluated on synthetic tasks, like Needle in a Haystack (Kamradt, 2023) test, which does not accurately represent real-world downstream task performance. In this work, we aim to create an open source long context recipe that poses the capability at the level of proprietary models (e.g., GPT-4-Turbo-2024-04-09) while focusing on real-world long-context understanding tasks.

On the other hand, the long-context capability of LLMs is sometimes considered a rival technique to retrieval-augmented generation (RAG). In fact, **RAG and long-context techniques complement each other from a pragmatic perspective**. An LLM with a long context window can either process large volumes of text as a prompt or utilize retrieval methods to efficiently extract relevant information from the extensive text, depending on the downstream tasks and accuracy vs. efficiency trade-offs. RAG has efficiency advantages and can easily retrieve relevant contexts for query-based tasks (e.g, QA) from billions of tokens, a feat that long context models cannot achieve so far. Meanwhile, long context models are good at tasks such as summarizing entire documents, where RAG can not perform. As a result, **the state-of-the-art (SOTA) LLM needs to excel at both capabilities, providing options for different downstream tasks and meeting different accuracy-efficiency trade-off requirements.** To compare the solution between long context and RAG, Xu et al. (2024) extended the context window of Llama2 (Touvron et al., 2023b) to 16K and 32K tokens, and studied the interplay between RAG and long-context LLMs. It demonstrates that the RAG method can improve the generation accuracy and inference efficiency of a GPT-3.5-turbo-16k level long context model for QA and query-based summarization tasks.

In this work, we present ChatQA 2, which extends the study of long-context vs RAG by pushing long-context LLMs to the GPT-4-Turbo level capability (with a 128K context window), and combines it with a state-of-the-art long-context retriever for RAG. See Table 1 for an overview of the results. The training data and reproduction recipe for this work are publicly accessible.

Specifically, we make the following contributions:

1. We present a two-step approach to establish the long context capability of Llama3-70B. First, we extend Llama3-70B base's context window from 8K to 128K by continually pretraining it on a mix of SlimPajama (Soboleva et al., 2023) with upsampled long sequences (Fu et al., 2024). Then, we apply a three-stage instruction tuning process on curated datasets to enhance the instruction-following, RAG, and long context understanding capabilities at each respective stage. We find that stage-wise instruction-tuning, by incorporating previous datasets, simplifies experimentation and hyperparameter tuning.

2. We demonstrate that the resulting `Llama3-ChatQA-2-70B` with 128K context window achieves better accuracy than most existing state-of-the-art models, including GPT-4-Turbo-2024-04-09, Qwen2-72B-Instruct, and Llama3.1-70B-Instruct, on both long-context tasks exceeding 100K tokens and the RAG benchmark for conversational QA tasks within a 4K context window.

3. Previous RAG approachs has two potential issues that can undermine downstream task accuracy: *i)* top-$k$ chunk-wise retrieval introduces fragmentation of context, and *ii)* a small top-$k$ leads to low recall, while a larger $k$ introduces too much irrelevant context to the LLM (e.g., see the analysis in Figure 1 of Yu et al., 2024). In contrast, we find that the RAG performance of our `Llama3-ChatQA-2-70B` is highly robust to variations in chunk size when using a long-context retriever (Wang et al., 2023c; Lee et al., 2024). Even more promising, accuracy consistently improves as the total number of retrieved tokens for input increases.

4. We conduct extensive comparisons between direct long-context and RAG solutions using the same state-of-the-art long-context LLMs. We find that RAG can still outperform GPT-4-Turbo-level long-context models (including our `Llama3-ChatQA-2-70B`) on both 32K benchmarks and real-world 128K tasks, given a sufficient number of top-k chunks.

We organize the rest of the paper as follows. We discuss related work in § 2. We introduce the continued pretraining for context window extension in § 3.1 and the three-stage instruction tuning in § 3.2. We report results in § 5 and conclude the paper in § 7.

## 2 RELATED WORK

### 2.1 LONG CONTEXT LLM

The trend of extending the context window in LLM starts by Claude with 100K token context (Anthropic, 2023). Although the underlying long context techniques behind proprietary models are unclear, the open LLM and research community has developed many methods to extend the context window of LLMs through continued training or fine-tuning (Kaiokendev, 2023; Nijkamp et al., 2023; Chen et al., 2023a; Tworkowski et al., 2023; Chen et al., 2023b; Peng et al., 2023; Xiong et al., 2023; Fu et al., 2024), especially for open-access LLMs (Touvron et al., 2023a;b) based on rotary position embedding (RoPE) (Su et al., 2024).

There are two popular approaches to adapt RoPE for long-context inputs: position interpolation (Chen et al., 2023a) and increasing the base frequency $\theta$ of RoPE (Xiong et al., 2023; Liu et al., 2023b). Recently, Yi-34B (01.AI et al., 2024) was pretrained with a sequence length of 4K, and its context window was extended to 200K by increasing the RoPE $\theta$ from 10,000 to 5M during continued pretraining. Qwen2-72B-Instruct (Alibaba-QWen, 2024) was trained on 32K-length contexts and extrapolated to a 128K context length using YaRN (Peng et al., 2023). Instead of extending the context window of the base model then applying instruction tuning, GradientAI (2024) directly fine-tunes the Llama-3-Instruct, which uses NTK-aware interpolation (Peng et al., 2023) and the formula from Liu et al. (2023b) to scale up $\theta$.

Note that, the Llama-3.1 models (Dubey et al., 2024) with a 128K context window were introduced after the release of this work. We add a comprehensive comparison between ChatQA-2 and Llama-3.1 in the experiments.

### 2.2 RETRIEVAL-AUGMENTED GENERATION (RAG)

Retrieval with a standalone retriever (e.g., Karpukhin et al., 2020; Wang et al., 2022; Lin et al., 2023; Lee et al., 2024) is a long-standing solution for handling long texts that cannot fit into the context window of language models. In previous work, various retrieval-augmented language models have been proposed (Nakano et al., 2021; Borgeaud et al., 2022; Wang et al., 2023b;a; Guu et al., 2020; Izacard & Grave, 2021; Izacard et al., 2022; Lewis et al., 2020; Huang et al., 2023; Khandelwal et al., 2019; Liu et al., 2024).

In general, the input to a retriever or dense embedding model is a chunk of text (i.e., a sequence of subword tokens), and the output is an embedding vector. These dense embedding vectors are then indexed and retrieved using k-nearest-neighbor retrieval. Previous dense-embedding-based retrievers only supported limited *chunk size* (e.g., 512 tokens) (e.g., Karpukhin et al., 2020; Wang et al., 2022; Lin et al., 2023) due to the context window size of pretrained BERT base model. In top-$k$ chunk-wise retrieval, the short chunk size increases context fragmentation. As a result, extending the context window of retrievers has become popular. For example, Jina Embeddings 2(Günther et al., 2023) and Nomic Embed (Nussbaum et al., 2024) support 8K tokens, while E5-mistral-7B (Wang et al., 2023c) and NV-Embed (Lee et al., 2024) support 32K tokens.

## 3 METHOD

### 3.1 EXTENDING CONTEXT WINDOW TO 128K

In this section, we present the method to extend the context window from 8K to 128K for Llama3. Note that, the Llama-3.1 models with 128K context window are released after this work. We compare our `Llama3-ChatQA-2-70B` and `Llama3-ChatQA-2-8B` with Llama-3.1-70B-Instruct and Llama-3.1-8B-Instruct in experiments.

We prepare our long context pretraining corpus from the Slimpajama (Soboleva et al., 2023) following Fu et al. (2024). We upsample long-context documents with the hyperparameter set as 0.1 to produce 10 billion tokens with sequence length of 128k. Since Llama3 is pretrained with a much higher RoPE base frequency of 500,000 compared to Llama2, we increased the RoPE base frequency to 150M accordingly. We set the batch size to 32 to fit 4 million tokens in a batch and use a learning rate of 3e-5 to train 2000 steps (8B tokens in total).

Interestingly, we found it more effective to separate different documents using special characters, such as "", rather than the reserved beginning and ending tokens <BOS> and <EOS>. We hypothesize that the <BOS> and <EOS> tokens in Llama3 signal the model to ignore previous chunks of text after pretraining, which is not helpful for the LLMs to adapt for longer context inputs.

### 3.2 INSTRUCTION-TUNING WITH LONG CONTEXT DATA

In this section, we present the instruction tuning method designed to enhance both long context understanding capability and RAG performance.

Specifically, we implement three stages of instruction-tuning. For the first two stages, we follow ChatQA 1.5 (Liu et al., 2024), where the model is initially trained on a high-quality instruction-following datasets, and then trained on a blend of conversational QA data with provided context. However, these two stages involve relatively short contexts, with a maximum sequence length of only 4K tokens. To enhance our model's capability to handle very long context sequences up to 128k tokens, we collect a long SFT dataset.

This dataset is collected through two categories: 1) For SFT data sequences less than 32k: We leverage existing long-context datasets from LongAlpaca12k, GPT-4 samples from Open Orca [1], and Long Data Collections [2]. 2) For sequence lengths between 32k and 128k: Since it is challenging to collect such SFT samples, we rely on synthetic datasets. We utilize NarrativeQA (Kočiský et al., 2018), [3] which includes summary paragraphs, questions, answers, and source long web pages. The summaries are human-generated based on the source web pages, while the question-answer pairs are annotated by humans using the summaries. To extend the context length, we inserted a summary into the corresponding long web page document at a random location, ensuring that sentence structure was not disrupted. This approach maintains the grounding of the question-answer pairs within the augmented long documents.. Since we used NarrativeQA for synthetic data generation, we intentionally excluded it from the evaluation benchmarks to avoid any potential data contamination. Both the full long SFT dataset and the short SFT dataset from the first two stages are then blended for training. We set the learning rate at 3e-5 and the batch size at 32.

### 3.3 LONG CONTEXT RETRIEVER MEETS LONG CONTEXT LLM

As we mentioned in previous section, the current RAG pipeline for LLM has the following issues: *i)* The top-*k* chunk-wise retrieval introduces non-negligible fragmentation of context for generating accurate answers. For example, previous state-of-the-art dense-embedding based retrievers (e.g., Li et al., 2023; Lin et al., 2023) only support 512 tokens. *ii)* Small top-*k* (e.g., 5 or 10) usually leads to relatively low recall, while much larger *k* (e.g., 100) can lead to worse generation (see Table 5 in Xu et al. (2024)) as the previous LLMs could not utilize too many chunked context very well (Liu et al., 2023a). To address the issue, we propose to use the most recent long-context retriever (Wang et al., 2023c; Lee et al., 2024), which can support thousands of tokens. In our setting, we use the E5-mistral

---

[1] https://huggingface.co/datasets/Open-Orca/OpenOrca
[2] https://huggingface.co/datasets/togethercomputer/Long-Data-Collections
[3] https://huggingface.co/datasets/deepmind/narrativeqa

embedding model (Wang et al., 2023c) as the retriever. The input of E5-Mistral model is a chunk of text represented by a sequence of subword tokens, and the output is an embedding vector. The long documents or corpus are chunked and embedded into a set of dense embedding vectors. Such dense embedding vectors are indexed and retrieved by k-nearest-neighbor retrieval at inference time, where the embedding of prompt / question is treated as query.

## 4 BASELINES AND EVALUATION BENCHMARKS

**Long context models** We compare our models against SOTA long context models: 1) GPT-4-Turbo-2024-04-09 (128K context window) (OpenAI, 2023), 2) Qwen2-72B-Instruct (128K context window) (Alibaba-QWen, 2024), 3) Llama-3-70B-Instruct-Gradient-262k (256K context window) (GradientAI, 2024), 4) Llama-3.1-8B-Instruct and Llama-3.1-70B-Instruct (128K context window). Note that, the Llama-3.1 models are released after this work.

**Retrieval-augmented generation (RAG)** We apply RAG to our ChatQA 2 models and baseline long-context LLMs when it is applicable to the downstream tasks, such as those with informative prompts or queries. By default, we use the E5-mistral retriever (Wang et al., 2023c), as it supports both short and long contexts. In our experiments, we use a chunk size of 1200 and select the top 5 chunks, as this configuration provides a good trade-off between cost and performance, as demonstrated in the ablation study in Section 6.

To give a comprehensive study of different context lengths, our evaluation benchmarks covers three categories, 1) ultra-long context benchmarks beyond 100K tokens, 2) long context benchmarks within 32K tokens, and 3) short context benchmarks within 4K tokens.

### 4.1 ULTRA-LONG CONTEXT BENCHMARKS BEYOND 100K TOKENS

InfiniteBench (Zhang et al., 2024b) is proposed to evaluate the ultra-long context capability of LLMs over 100K sequence length. As we focus on real-world english tasks, we only take the four related tasks from the InfiniteBench, i.e. longbook summarization (En.Sum), longbook qa (En.QA), longbook multiple choice (En.MC), and longbook dialogue (En.Dia). En.Sum is a task that requires models to generate a concise summary of the given novel and is evaluated using the ROUGE-L-Sum metric (Lin, 2004). En.QA is annotated by a pipeline that ensures the questions' necessitating of long-range dependencies and reasoning, beyond simple short passage retrieval. Aggregation reasoning and filtering reasoning are the two primary reasoning categories. F1 score is used to evaluate the quality of the answer. En.MC is annotated with the same pipeline of En.QA except that four answer choices are provided and exact matching scores are reported. En.Dia leverages movie and drama scripts from a designated online database [4] with long, multi-role dialogues. Exact matching score is used again to evaluate the prediction accuracy.

### 4.2 LONG CONTEXT BENCHMARKS WITHIN 32K TOKENS

We use the long context datasets (except NarrativeQA as it is included in our training) from Xu et al. (2024) as our benchmark for long datasets within 32K. There are six datasets in total, where QMSum (QM), Qasper (QASP), QuALITY (QLTY) are token from SCROLLS (Shaham et al., 2022) and HotpotQA (HQA) MuSiQue (MSQ), MultiFieldQA-en (MFQA) are token from LongBench (Bai et al., 2023). Following the official metrics, we report the geometric mean of ROUGE scores (i.e., ROUGE1/2/L) (Lin, 2004) for QM, the exact matching (EM) score for QLTY, and F1 scores for the remaining four datasets QASP, MSQ, HQA and MFQA.

### 4.3 SHORT CONTEXT WITHIN 4K TOKENS

We use ChatRAG Bench (Liu et al., 2024) as our benchmark for short context within 4k. ChatRAG bench consists of 10 datasets and we exclude HDial as it is included in our training. Following the setup of Liu et al. (2024), for Doc2Dial (D2D), QuAC, and QReCC task with long documents, each document is divided into segments of roughly 300 words. The top 5 relevant chunks are then retrieved

---

[4]https://imsdb.com

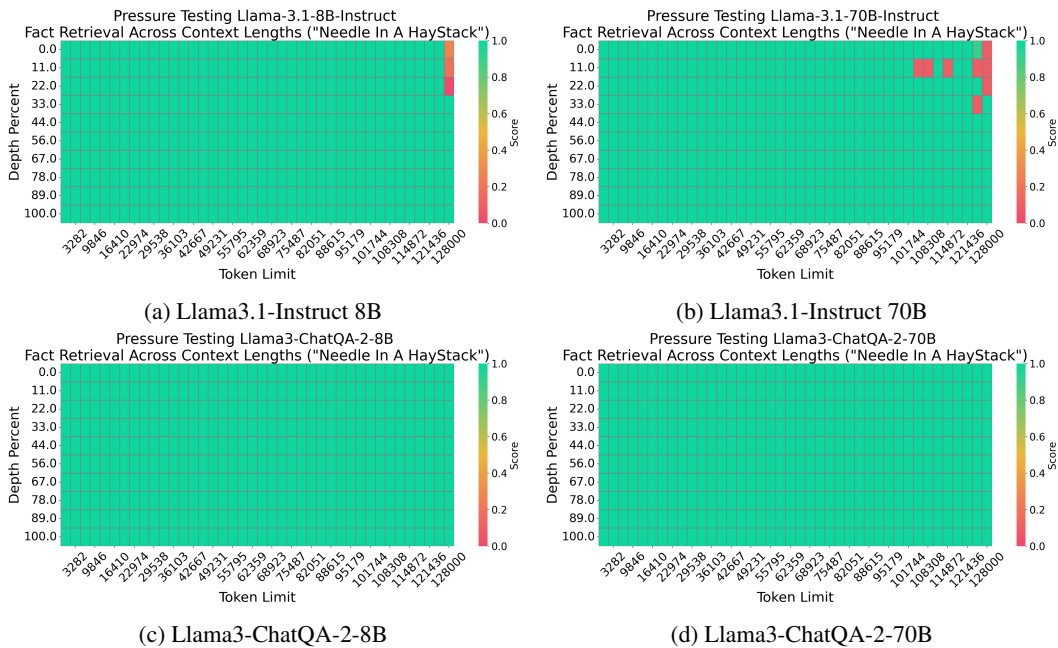

(a) Llama3.1-Instruct 8B

(b) Llama3.1-Instruct 70B

(c) Llama3-ChatQA-2-8B

(d) Llama3-ChatQA-2-70B

Figure 1: Needle In A Haystack test for (a) Llama3.1-Instruct 8B, (b) Llama3.1-Instruct-70B, (c) Llama3-ChatQA-2-8B, and (d) Llama3-ChatQA-2-70B, up to 128K context window. We show the result using the same needle: *"The best thing to do in San Francisco is eat a sandwich and sit in Dolores Park on a sunny day."*

as context for each user question. For TopiOCQA and INSCIT, top-20 chunks were retrieved to obtain similar context length to the first three datasets. The other four datasets are CoQA, DoQA, ConvFinQA (CFQA), and SQA, which cover a wide range of domains like finance, children's stories, literature, mid/high school exams, news, Wikipedia and etc. We use F1 score as the metric to evaluate the generations and report the average score without HDial as it is a fair zero-shot comparisons over different models.

## 5 RESULTS

In this section, we present the results and comparisons from extensive benchmark evaluations. We begin with the synthetic Needle in a Haystack test, then focus on real-world long context understanding and RAG tasks.

### 5.1 NEEDLE IN A HAYSTACK

We evaluate our `Llama3-ChatQA-2-70B` and `Llama3-ChatQA-2-8B` model on the Needle In A Haystack (NIAH) test (Kamradt, 2023). This synthetic task is popular for testing the long-context capability of LLMs, and can be considered as a threshold level evaluation. We use the needle test: *"The best thing to do in San Francisco is eat a sandwich and sit in Dolores Park on a sunny day."* We find this is generally a harder pressure test than the "passkey" in a haystack test, which is commonly used by other long-context LLM works. Figure 3 demonstrates the performance of our models up to 128K tokens. It shows that our model achieves 100% accuracy under both test cases. This test confirms our model's excellent long-context retrieval capability.

In addition, we also include the NIAH test for Llama3.1-Instruct 8B and Llama3.1-Instruct-70B. From Figures 1a and 1b, we find that neither model could pass the NIAH test with the needle: *"The best thing to do in San Francisco..."*, although they could pass the easier *Passkey* retrieval test, as shown in Figures 4a and 4b in Appendix.

| Model | Avg. | En.Sum | En.QA | En.MC | En.Dia |
|---|---|---|---|---|---|
| GPT-4-Turbo-2024-04-09 | 33.16 | 17.62 | 19.29 | 77.73 | 18.00 |
| Claude 2 | **33.96** | 14.50 | 11.97 | 62.88 | 46.50 |
| Yi-34B-200K | < 15.15 | < 5 | 12.17 | 38.43 | < 5 |
| w/ RAG | N/A | N/A | 17.69 | 77.29 | N/A |
| Qwen2-72B-Instruct | **39.77** | 31.66 | 21.46 | 82.97 | 23.00 |
| w/ RAG | N/A | N/A | 16.48 | 76.86 | N/A |
| Llama-3-70B-Instruct-Gradient-262k | 32.57 | 14.27 | 29.52 | 69.00 | 17.50 |
| Llama3.1-8B-Instruct | 33.17 | 29.20 | 31.51 | 58.95 | 13.00 |
| w/ RAG | N/A | N/A | 15.41 | 64.63 | N/A |
| Llama3.1-70B-Instruct | **39.81** | 30.94 | 38.49 | 75.55 | 14.29 |
| w/ RAG | N/A | N/A | 23.44 | 79.04 | N/A |
| Llama3-ChatQA-2-8B | 35.59 | 17.14 | 43.51 | 64.19 | 17.50 |
| w/ RAG | N/A | N/A | 41.48 | 63.76 | N/A |
| Llama3-ChatQA-2-70B | **41.04** | 16.08 | 48.22 | 80.35 | 19.50 |
| w/ RAG | N/A | N/A | 40.66 | 73.80 | N/A |

Table 2: Evaluation results on InfiniteBench includes real-world long-context understanding tasks beyond a 100K context window. For RAG, we use top-5 retrieved chunks, each with 1200 tokens from E5-mistral retriever (Wang et al., 2023c).

## 5.2 ULTRA-LONG CONTEXT EVALUATION BEYOND 100K TOKENS

In this subsection, we evaluate the ultra-long context capability beyond 100K tokens on the real-world tasks from InfiniteBench (Zhang et al., 2024a). Table 2 shows that our model (41.04) outperforms many existing state-of-the-art models, such as GPT4-Turbo-2024-04-09 (33.16), Llama3.1-70B-Instruct (39.81), Qwen2-72B-Instruct (39.77), Llama-3-70B-Instruct-Gradient-262k (32.57) and Claude 2 (33.96), confirming the effectiveness of our recipe and state-of-the-art long-context capability of our model. We find that our model performs extremely well at QA tasks while performs relatively low for summarization. This can be attributed to the lack of summarization data in our SFT recipe. Additionally, we compare Llama3-ChatQA-2-8B to Llama3.1-8B-Instruct and our model again shows better results. The RAG setting shows it is worse than the full long context scores, when we employed the default RAG evaluation setting: top-5 retrieved chunks with a chunk size of 1200 tokens, so overall $5 \times 1,200 = 6,000$ are fed into LLMs. In Section 6 and Table 5, we show that RAG scores can be better than full long context solutions if we increase the number of chunks in the prompt.

## 5.3 LONG CONTEXT EVALUATION WITHIN 32K TOKENS

In this subsection, we evaluate the long context capability within 32K tokens. Table 3 shows that GPT-4-Turbo-2024-04-09 achieves the highest score of 51.93 among all models. Our model scores 48.15, which is higher than Llama-3-70B-Instruct-Gradient-262k but is a bit lower than Qwen2-72B-Instruct (49.94) and Llama3.1-70B-Instruct (49.92). This difference can be attributed to the extensive 32K pretraining and large SFT data implemented by Qwen2-72B-Instruct and Llama3.1-70B-Instruct, while we used a much smaller continued pretraining corpus and SFT datasets.

Additionally, we found that the a default RAG solution (top-5 with chunk size as 1200) perform worse than the top long context solution (e.g. GPT-4-Turbo-2024-04-09, Qwen2-72B-Instruct, Llama3-ChatQA-2-70B), which suggests that SOTA long context LLMs can handle 32K tokens well within their context window.

## 5.4 CHATRAG BENCH: SHORT CONTEXT EVALUATION WITHIN 4K TOKENS

In this subsection, we evaluate the models on the short context tasks within 4K tokens from CHATRAG BENCH (Liu et al., 2024). The results can be found in Table 4. Our model achieves an average score of 56.30, outperforming most existing SOTA models with 128K context window, including GPT-4-Turbo-2024-04-09, Qwen2-72B-Instruct, and Llama3.1-70B-Instruct. It is also comparable, though slightly behind, Llama3-ChatQA-1.5-70B (57.14), which only supports 4K context length. This suggests that extending short-context models to long-context has trade-offs. Exploring how to effectively extend the context window to even larger scales, e.g., millions of tokens as in Gemini

| Model | Avg. | QM | QASP | QLTY | MSQ | HQA | MFQA |
|---|---|---|---|---|---|---|---|
| GPT-4-Turbo-2024-04-09 | **51.93** | 16.37 | 38.96 | 88.45 | 44.88 | 70.65 | 52.26 |
| w/ RAG | 49.84 | 16.07 | 36.18 | 85.85 | 42.17 | 67.85 | 50.94 |
| Qwen2-72B-Instruct | **49.94** | 17.06 | 34.84 | 84.85 | 46.80 | 65.98 | 50.12 |
| w/ RAG | 48.08 | 17.63 | 35.19 | 83.05 | 39.92 | 64.58 | 48.10 |
| Llama-3-70B-Instruct-Gradient-262k | 40.51 | 20.72 | 30.64 | 74.35 | 20.20 | 45.82 | 51.33 |
| w/ RAG | 40.57 | 20.04 | 30.68 | 72.35 | 22.19 | 46.85 | 51.31 |
| Llama3.1-8B-Instruct | 42.42 | 15.16 | 36.59 | 68.20 | 25.89 | 52.42 | 56.25 |
| w/ RAG | 41.35 | 14.08 | 35.05 | 65.35 | 28.30 | 50.76 | 54.54 |
| Llama3.1-70B-Instruct | **49.92** | 14.85 | 35.75 | 87.40 | 45.65 | 59.64 | 56.23 |
| w/ RAG | 47.63 | 13.59 | 32.11 | 85.65 | 39.53 | 58.30 | 56.58 |
| Llama3-ChatQA-2-8B | 39.41 | 11.64 | 28.85 | 63.35 | 27.81 | 53.81 | 51.02 |
| w/ RAG | 40.31 | 13.20 | 28.85 | 61.10 | 29.77 | 57.81 | 51.15 |
| Llama3-ChatQA-2-70B | **48.15** | 15.99 | 32.84 | 82.20 | 39.26 | 62.24 | 56.39 |
| w/ RAG | 47.59 | 16.16 | 33.27 | 81.70 | 38.16 | 61.13 | 55.14 |

Table 3: Evaluation results on the long context benchmarks within 32K tokens. For RAG, we use top-5 retrieved chunks, each with 1200 tokens from E5-mistral retriever (Wang et al., 2023c). Note that here, we do not intend to optimize the RAG scores by increasing the number of total retrieved tokens and we leave the ablation in Section 6.

| Models | Avg. w/o HDial | D2D | QuAC | QReCC | CoQA | DoQA | CFQA | SQA | TCQA | INSCIT |
|---|---|---|---|---|---|---|---|---|---|---|
| GPT-3.5-Turbo-0613 | 50.69 | 34.83 | 37.17 | 50.46 | 79.33 | 41.11 | 73.15 | 60.63 | 44.30 | 35.27 |
| GPT-4-0613 | 54.35 | 34.16 | 40.29 | 52.01 | 77.42 | 43.39 | 81.28 | 79.21 | 45.09 | 36.34 |
| GPT-4-Turbo-2024-04-09 | **54.72** | 35.35 | 40.10 | 51.46 | 77.73 | 41.60 | 84.16 | 79.98 | 48.32 | 33.75 |
| Command R+ 104B | 51.40 | 33.51 | 34.16 | 49.77 | 69.71 | 40.67 | 71.21 | 74.07 | 53.77 | 35.76 |
| Qwen2-72B-Instruct | **54.06** | 35.76 | 38.48 | 51.21 | 85.04 | 33.89 | 77.52 | 77.06 | 51.64 | 35.90 |
| Llama2-Chat-70B | 44.64 | 36.87 | 32.47 | 49.40 | 80.41 | 38.97 | 46.85 | 37.62 | 44.31 | 34.88 |
| Llama3-Instruct-70B | 52.95 | 37.88 | 36.96 | 51.34 | 76.98 | 41.24 | 76.60 | 69.61 | 49.72 | 36.23 |
| Llama3-70B-Instruct-Gradient-262k | 45.20 | 34.30 | 24.01 | 49.60 | 73.45 | 25.76 | 54.70 | 46.30 | 34.89 |
| Llama3-ChatQA-1.5-70B | **57.14** | 41.26 | 38.82 | 51.40 | 78.44 | 50.76 | 81.88 | 83.82 | 55.63 | 32.31 |
| Llama3.1-8B-Instruct | 48.79 | 35.24 | 33.03 | 50.59 | 74.32 | 29.84 | 69.59 | 68.18 | 45.49 | 33.85 |
| Llama3.1-70B-Instruct | 52.12 | 35.80 | 33.92 | 51.30 | 73.46 | 34.14 | 77.72 | 77.45 | 50.62 | 34.69 |
| Llama3-ChatQA-2-8B | 52.50 | 39.04 | 37.85 | 44.76 | 76.14 | 49.3 | 75.37 | 68.25 | 51.76 | 29.99 |
| Llama3-ChatQA-2-70B | **56.30** | 40.28 | 39.95 | 46.32 | 81.38 | 52.08 | 79.60 | 79.40 | 56.51 | 31.17 |

Table 4: Evaluation results on CHATRAG BENCH with 9 datasets. Following Liu et al. (2024), we exclude HDial as it is included in the instruction tuning datasets. The maximum context lengths are 4K tokens.

| | RAG (top-$k$) | Long context |
|---|---|---|
| Llama3-ChatQA-2-70B | **64.55** ($k$=30) | 64.29 |
| Qwen2-72B-Instruct | **52.95** ($k$=20) | 52.22 |

Table 5: We compare RAG vs. long context evaluation using our Llama3-ChatQA-2-70B on tasks beyond 100k. Here we use average accuracy of En.QA and En.MC tasks that can apply RAG. The RAG-based result is still better than long context evaluation.

1.5 Pro (Gemini-Team, 2024), without degrading performance on regular short-context tasks is an exciting research direction.

## 5.5 ABLATION STUDY

| Retriever | Long (32K) | En.MC | En.QA |
|---|---|---|---|
| E5-Mistral | 47.59 | 73.80 | 40.66 |
| NV-emb-v2 | 46.43 | 74.24 | 41.19 |

Table 6: Ablation of different retrievers for Llama3-ChatQA-2-70B using the top-5 and chunk size as 1200.

To understand the choice of different retrievers, we also report the performance of using NV-emb-v2 for Llama3-ChatQA-2-70B, the top-1 retriever in the MTEB leaderboard [5]. Table 6 shows (i) there

---

[5]https://huggingface.co/spaces/mteb/leaderboard

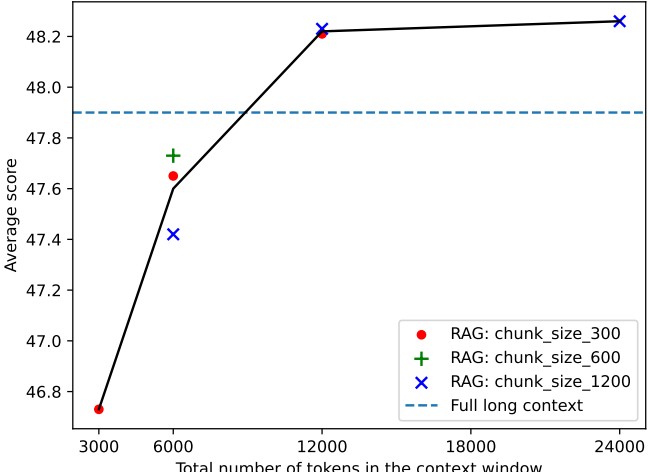

Figure 2: Ablation of `Llama3-ChatQA-2-70B` with RAG given different top-$k$ = {5, 10, 20, 40} retrieval, and chunk-size = {300, 600, 1200} on long context benchmarks within 32K tokens (see Section 4.2 for more details). The accuracy can be monotonically improved with more retrieved tokens (i.e., $k\times$ chunk_size) in the context window.

are relatively small performance variations when switching to another high-performing retriever and (ii) there is no clear overall winner, with NV-Embed-v2 showing better performance on ultra-long tasks.

|  | All-in-one-stage | Three-stage |
|---|---|---|
| Short (4K) | 48.55 | 52.50 |
| Long (32K) | 40.69 | 39.41 |
| Ultra-long (>100K) | 35.07 | 35.59 |

Table 7: Comparison between the three-stage training and all-in-one-stage training.

To study the effect of three stage recipe, we blended all the instruction-tuning data from all three stages and trained a new model based on Llama-3-8B base in a single stage. The results are shown in Table 10: we observed that the three-stage training process outperforms the all-in-one stage training on RAG by a significant margin, while showing marginally better or worse performance on ultra-long and long tasks.

To further improve the performance of our long (32K) setting, we collect another SFT data which contains 1.6M samples covering a wide range topics. Table 8 shows that our `Llama3-ChatQA-2-8B` (new) is comparable to Llama3.1-8B-Instruct on Long (32K) tasks while being significantly better at Short (4K) and Ultra-long (>100K) tasks. Furthermore, it shows that (i) our model outperforms Llama3.1-8B-Instruct on the math benchmark GSM8K (ii) it underperforms on the knowledge-intensive MMLU and the coding benchmark HumanEval. Since it has not been aligned by DPO / PPO process as Llama3.1-8B-Instruct, the MT-Bench score is lower. We could apply the RLHF process to improve its MT-Bench score in future.

## 6 RAG VERSUS LONG CONTEXT

In Figure 2 and Table 5, we compare direct long-context and RAG solutions on 32K and beyond 100K benchmarks. For sequences beyond 100K tokens, we report only the average scores for En.QA and En.MC, as the RAG setting is not directly applicable to En.Sum and En.Dia.

|  | Llama3-ChatQA-2-8B (new) | Llama3.1-8B-Instruct |
|---|---|---|
| Short (ChatRAG) | **53.79** | 48.79 |
| Long (32K) | 42.05 | **42.42** |
| Ultra long (>100K) | **35.18** | 33.17 |
| HumanEval | 66.46 | **70.73** |
| MMLU | 65.73 | **67.59** |
| MT-bench | 8.09 | **8.42** |
| GSM8K (0-shot) | **87.41** | 83.70 |

Table 8: Comparison between Llama3.1-8B-Instruct and a new `Llama3-ChatQA-2-70B` model trained by 1.6M SFT dataset

Figure 2 compares different chunk sizes for top-$k$ retrieval and the total number of tokens fed into our long context LLM. Comparing total tokens from 3000 to 24000, we found that more tokens consistently yield better results, confirming the strong long-context capability of our model. Also, for downstream tasks with sequence lengths up to 32K, RAG can still marginally outperform the long-context solution if a sufficient number of top-$k$ chunks are used, i.e., at least 12,000 tokens as shown in Figure 2. Therefore, the hyperparameter $k$ can be selected based on the tradeoff between efficiency and accuracy in real-world applications.

For context lengths beyond 100K in Table 5, RAG (using top-30 chunks for our `Llama3-ChatQA-2-70B`, and top-20 for Qwen2-72B-Instruct) again outperforms the full long-context solution. In such scenarios, RAG is recommended for better accuracy and much lower inference cost, provided it is applicable to the downstream tasks.

## 7    CONCLUSION

We introduce `Llama3-ChatQA-2-70B`, a long context LLM that possess GPT-4 Turbo-level capabilities for both understanding up to 128K long contexts and utilizing retrieved contexts for generation. This provides the flexible options for different downstream tasks with specific accuracy and efficiency requirements. We present a detailed and reproducible technical recipe for building and evaluating the model, including the methods, training data, and evaluation benchmarks. In particular, we evaluate ChatQA 2 on short-context RAG benchmark (ChatRAG) (within 4K tokens), long context tasks from SCROLLS and LongBench (within 32K tokens), and ultra-long context tasks from InfiniteBench (beyond 100K tokens). Our results demonstrate that the `Llama3-ChatQA-2-70B` can achieve GPT-4-Turbo-2024-04-09 level accuracy on these benchmarks across different sequence lengths.

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

# A  APPENDIX

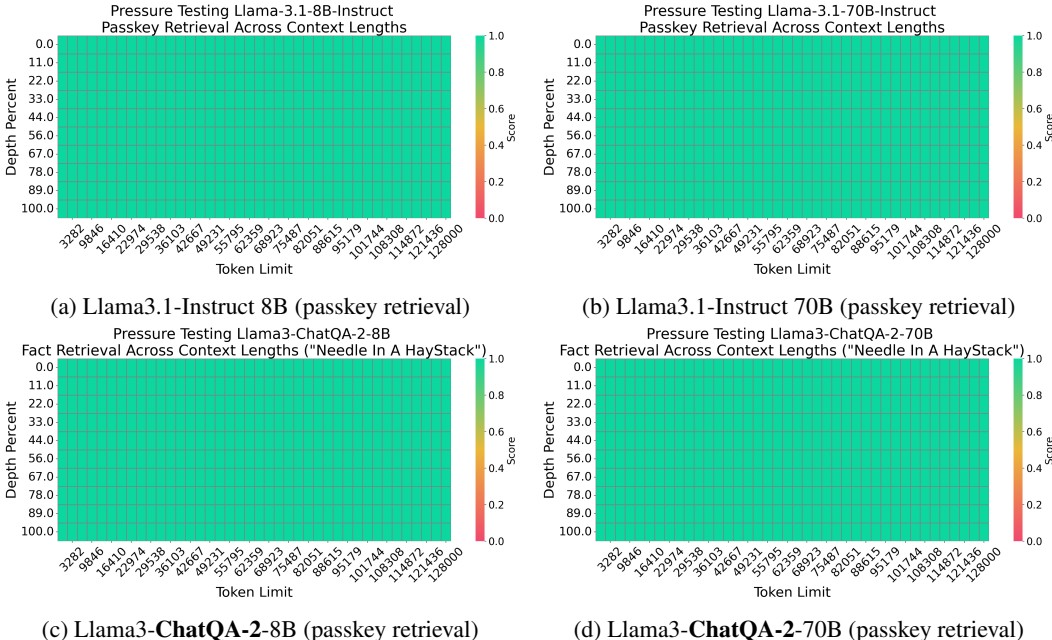

(a) Llama3.1-Instruct 8B (passkey retrieval)     (b) Llama3.1-Instruct 70B (passkey retrieval)

(c) Llama3-**ChatQA-2**-8B (passkey retrieval)     (d) Llama3-**ChatQA-2**-70B (passkey retrieval)

Figure 3: Needle in A Haystack test for Llama3.1-Instruct 8B and Llama3.1-Instruct-70B up to 128K context window. We show the *Passkey* retrieval results here. The needle is set as *"The pass key is 385243. Remember it. 385243 is the pass key."* with the question asking *"What is the pass key?"*

To further confirm the superrior retrieval performance, we evaluate our models on the three retrieval based tasks in InfiniteBench in Table 9. Aligned with our superior NIAH scores, all our models get 100% accuracy for number_string and passkey tasks and our `Llama3-ChatQA-2-70B` get a better score than GPT-4-Turbo on kv_retrieval task.

| model | kv_retrieval | numbe_string | passkey |
|---|---|---|---|
| GPT-4-Turbo | 89 | 100 | 100 |
| Llama3-ChatQA-2-70B | 91 | 100 | 100 |
| Llama3-ChatQA-2-8B | 72 | 100 | 100 |

Table 9: `Llama3-ChatQA-2-70B` demonstrates better performance on three retrieval tasks in InfiniteBench.

# B  APPENDIX

We trained our model on 2B tokens with different separators and the NIAH below shows that using document breaker "" is much better than <EOS> <BOS>.

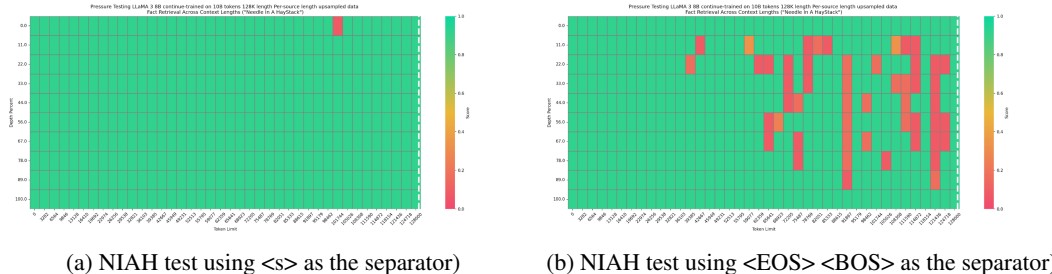

(a) NIAH test using  as the separator)                    (b) NIAH test using <EOS> <BOS> as the separator)

Figure 4: NIAH shows that using document breaker "" is much better than <EOS> <BOS>

## C  APPENDIX

The three-stage SFT process involves blending multiple datasets, with each dataset potentially being trained for a different number of epochs, depending on its blending ratio and number of samples. For instance, the statistics for the datasets used in Stage 2 are provided below, which adds complexity to the analysis.

|  | number of samples | epochs |
|---|---|---|
| drop | 29195 | 0.68 |
| Narrative_QA | 39440 | 0.69 |
| Quoref | 10996 | 0.68 |
| ROPES | 10924 | 0.69 |
| squad1.1 | 86863 | 0.32 |
| squad2.0 | 86092 | 0.32 |
| newsqa | 76560 | 0.36 |
| convqa | 5413 | 3.73 |
| nvolve_qa | 55458 | 1.20 |
| scale_cqa | 18224 | 2.37 |
| instruct_v2 | 1411 | 4.50 |
| hybriddial | 19190 | 2.70 |
| convfinqa | 1162 | 6.70 |
| tatqa | 3176 | 5.45 |
| corner_cases | 1720 | 4.19 |

Table 10: Comparison between the three-stage training and all-in-one-stage training.

