# OpenReview forum: "ChatQA 2: Bridging the Gap to Proprietary LLMs in Long Context and RAG Capabilities"
_ICLR.cc/2025/Conference — ICLR 2025 Poster_

### Official Review · Reviewer_GdeC · 2024-10-18

**Soundness:** 4
**Presentation:** 1
**Contribution:** 4
**Rating:** 6
**Confidence:** 2

**Summary:**

This paper presents Llama3-ChatQA-2, a model based on LLama-3-base that is trained with an extended context window. The authors first collect a long SFT dataset for instruction-tuning and train the model with modified PE. On top of this, they integrate a long-context retriever to implement RAG. Extensive testing are conducted across a variety of tasks involving short, long, and ultra-long token lengths, achieving state-of-the-art (SOTA) results.

**Strengths:**

1. The paper provides open access to the trained model and training recipe, which facilitates further research in the field of LLM + RAG.
2. The paper focuses on increasing the context length of LLMs and has conducted extensive testing across benchmarks of varying lengths. It compares the performance with both open-source and proprietary models, achieving promising results.

**Weaknesses:**

Some sections of the paper are too brief, presuming the reader’s familiarity with certain concepts, particularly those related to RAG. For example, the meaning of “chunk size” and a simple introduction to the inputs and outputs of the E5-mistral retriever are not sufficiently explained. Since the paper emphasizes *open-source training data and reproduction recipe* compared to the open-access (Open Weights) Llama3.1-70B-Instruct, it would benefit from providing more detailed explanations, like [1].

> [1] The llama 3 herd of models. *arXiv preprint arXiv:2407.21783*.

**Questions:**

1. Under what circumstances does RAG improve LLM performance? For instance, why does RAG cause a performance drop for Llama3-ChatQA-2 on En.MC in Table 2, while improving the performance of Llama3.1-Instruct? The SOTA results in Tables 2 and 3 were achieved without RAG. Does this suggest that the effectiveness of RAG depends on the specific benchmark rather than the context length?
2. Could you test the results with retrievers other than E5-mistral?

---

> ### Author Response · Authors · 2024-11-27
>
> Many thanks for your detailed comments and helpful feedback. They are helpful to improve the quality of our work. We will address your comments in the following:
>
> ---
>
> > “Some sections of the paper are too brief, presuming the reader’s familiarity with certain concepts, particularly those related to RAG. For example, the meaning of “chunk size” and a simple introduction to the inputs and outputs of the E5-mistral retriever are not sufficiently explained. Since the paper emphasizes open-source training data and reproduction recipe compared to the open-access (Open Weights) Llama3.1-70B-Instruct, it would benefit from providing more detailed explanations”
> - We sincerely appreciate your feedback. The chunk size refers to the number of tokens or subwords in each chunk, representing the length of the input sequence for a retriever or embedding model. The E5-Mistral retriever processes a sequence of tokens as input and produces an embedding vector as output. These dense embedding vectors are then indexed and retrieved using k-nearest-neighbor retrieval.
> - In Section 2.2, we have added more background information on RAG in the updated draft, including an explanation of “chunk size.” Additionally, in Section 3.3, we provide more detailed explanations related to E5-Mistral retriever.
>
> ---
>
> > “Under what circumstances does RAG improve LLM performance? For instance, why does RAG cause a performance drop for Llama3-ChatQA-2 on En.MC in Table 2, while improving the performance of Llama3.1-Instruct? The SOTA results in Tables 2 and 3 were achieved without RAG. Does this suggest that the effectiveness of RAG depends on the specific benchmark rather than the context length?”
> - This is a good question. First, the SOTA results in Tables 2 and 3 were achieved without using RAG, as we employed the default RAG evaluation setting: top-5 retrieved chunks with a chunk size of 1,200 tokens, so overall $5 \times 1,200=6,000$ are fed into LLMs. In Figure 2 and Table 5, we demonstrate that providing more retrieved tokens as input (e.g., by increasing top-k and chunk size) to Llama3-ChatQA-2-72B consistently improves the scores, e.g. from 6,000 to 24,000 tokens. As a result, RAG can match or even surpass the full long-context LLM results if one is willing to trade inference efficiency for improved accuracy.
> - The En.MC task consists of 229 multiple-choice questions, so the accuracy may show some variations. In such cases, the average score across multiple benchmarks provides a more reliable measure than a single benchmark.
>
> ---
>
> > “Could you test the results with retrievers other than E5-mistral?”
> - Many thanks for your suggestion. We evaluated the state-of-the-art retriever, NV-Embed-v2, and the results for our Llama3-ChatQA-2-70B model are provided below. The comparison highlights: (i) relatively small performance variations when switching to another high-performing retriever and (ii) no clear overall winner, with NV-Embed-v2 showing better performance on ultra-long tasks.
>
> | Retriever | Long tasks (32K)  | longbook_choice |  longbook_qa  |
> | -------- | ------- | ------- | ------- |
> | E5-Mistral | 47.59 | 73.8 | 40.66 |
> | NV-emb-v2 | 46.43 | 74.24 | 41.19 |
>
> ---
> ---
>
> Many thanks again for your valuable feedback and helpful suggestions. We hope our response helps to address your concerns. If you have any additional questions, please let us know. We would be happy to discuss them further.

---

> > ### Comment · Reviewer_GdeC · 2024-12-01
> >
> > Thank you for the author’s response, which has partially resolved my problems and concerns. I have decided to keep my rating unchanged and hope the AC will consider my confidence level when making the final decision, as I am not very familiar with this field.

---

> > > ### Author Response · Authors · 2024-12-02
> > >
> > > Dear Reviewer,
> > >
> > > Thank you so much for your kind reply. Your suggestions for improving the presentation are particularly helpful; we have updated the paper accordingly. We are glad to know that our response has helped address some of your questions and concerns. If you have any additional questions, please feel free to reach out. We would be happy to discuss further.
> > >
> > > Best,
> > >
> > > Authors

---

### Official Review · Reviewer_TcQ9 · 2024-11-04

**Soundness:** 3
**Presentation:** 4
**Contribution:** 3
**Rating:** 6
**Confidence:** 4

**Summary:**

This paper introduces a recipe to extend Llama-3.0 models to long context windows (128K). The recipe involves increasing the RoPE frequency, continual pretraining on longer length corpus, then supervised fine-tuning on a mix of short and long context length instructions. They benchmark the model on tasks (synthetic and real-world) with various context lengths and found consistent gains over the original Llama-3 models, with the 70B model exceeding the performance of GPT-4-Turbo on some tasks. Ablation studies demonstrate tradeoffs involved with extending LLMs to handle long-context inputs.

**Strengths:**

## Originality
The technique of extending the context lengths of LLMs is not new. The most originality lies in the creation of the synthetic, long-context instruction dataset highlighted in Section 3.2. The authors seem to suggest that utilizing long-context retrievers for RAG is a contribution in Section 3.3.

## Quality
The paper is well written. Experiments cover a wide range of tasks and reflect actual usage of long-context LLMs in real life. Baselines covered are extensive. Nothing to pick here.

## Clarity
The recipe is simple and presented clearly in the paper. Particularly, technical details are well-documented in the Method Section (Sec 3).

## Significance
The topic of extending LLMs to support longer context than it was original pretrained is of itself significant, which in turn makes potential solutions significant.

**Weaknesses:**

## Discussion on the key factors influencing performance tradeoff amongst tasks with different context lengths in long-context models

A major part of the paper is dedicated to compare the performance of long-context models on various (shorter) context lengths, specifically 4K, 32K, and 100K+. As the authors stated in P8L429 "This suggests that extending short-context models to long-context has trade-offs.", it is important to pinpoint what exactly controls the trade-off. The authors claimed in P7L372 that "This difference (in 32K performance) can be attributed to the extensive 32K pretraining and large SFT data implemented by Qwen2-72B-Instruct and Llama3.1-70B-Instruct.", which seems to hint at the data distribution being mainly responsible for the tradeoff. However, to the extent of my knowledge, Llama3.1 didn't released their training recipe nor data distribution. Can the tradeoff be fully attributed to the context-length distribution in the continual pretraining and SFT dataset? More discussion would greatly benefit the work.

## Relevance of stage-wise instruction-tuning

The authors highlighted in P2L98 and P4L187 that the instruction-tuning is split into 3 stages: instruction-following on short context, QA on short context, then a mix of long context instruction-following and synthetic QA. Stated in P4L199 "Both the full long SFT dataset and the short SFT dataset from the first two stages are then blended for training." What is the relevance of breaking down the instruction-tuning into stages when the final stage includes all data from the first two stages? An ablation study on training the model all-at-once with all the available data would help justify the multi-stage approach. Perhaps there are benefits in gradually increasing the context-length when instruction-finetuning and is actually key to the success of this recipe.

## Evaluating tradeoffs with other LLM capabilities with standard, general benchmarks

Section 5.2 highlights the difference in performance of the proposed model on QA and summarization. Quote "We find that our model performs extremely well at QA tasks while performs relatively low for summarization. This can be attributed to the lack of summarization data in our SFT recipe." Indeed the SFT data is heavy on QA and thus accounting for the performance bias. It is concerning that the good performance on this model on the benchmarked tasks might merely be a tradeoff with other LLM capabilities (e.g. reasoning, ICL, commonsense, creative writing, coding). Providing results on more general benchmarks (e.g. MT-Bench, MMLU, HumanEval) would help users evaluate the entire spectrum of tradeoffs of the proposed recipe.

**Questions:**

See weakness section for questions.

---

> ### Author Response · Authors · 2024-11-27
>
> Many thanks for your comments and feedback. We will address your comments in the following:
>
> ---
>
> > “Discussion on the key factors influencing performance tradeoff amongst tasks with different context lengths in long-context models …”
>
> - This is an excellent question. Our observation about "extending short-context models to long-context" having trade-offs comes from the comparison between ChatQA-2-128K and ChatQA-1.5-8K, which only supports 8K context length. ChatQA-2 shares the same 1st and 2nd stages of SFT as ChatQA-1.5, allowing it to achieve a ChatRAG score (4K context length) very comparable to ChatQA-1.5. Note that, the 1st and 2nd stages of SFT datasets are all short context i.e., within 4K length. However, after the 3rd stage of long-context SFT with a lot of long context datasets (up to 128K length), ChatQA-2's ChatRAG score degraded from 57.14 to 56.30, while achieving impressive performance on 100K and 32K benchmarks. This suggests the context-length distribution of SFT datasets plays a role in such tradeoffs.
>
> - As you mentioned, both Llama3.1 and Qwen2 didn’t release their training recipe nor data distribution, making it infeasible to make comparison. We hypothesized that the primary reason ChatQA 2 underperforms compared to Llama 3.1 on long-context benchmarks within 32K tokens is the limited availability of high-quality open-source SFT datasets, especially when compared to the proprietary SFT datasets used in Llama 3.1 and Qwen2. As a result, we tried to expand the amount of SFT data used in ChatQA 2 training. During the rebuttal phase, we trained another 8B ChatQA 2 model by expanding the first-stage SFT dataset from 128K samples to 1.6M samples. This resulted in an improvement in the score on long-context tasks (within 32K tokens context length) from 39.41 to 42.05, bringing it very close to Llama3.1-8B-Instruct. Please see the table below for details. Our findings indicate that increasing the amount of SFT data plays a crucial role in improving performance.
>
> | model | Llama3-ChatQA-2-8B   | Llama3.1-8B-Instruct |  Llama3-ChatQA-2-8B (1.6M 1st stage SFT samples)  |
> | -------- | ------- | ------- | ------- |
> | scores  | 39.41   | 42.42  |  42.05  |
>
> ---
> ---
>
> > “Relevance of stage-wise instruction-tuning ...”
>
> - **Ablation of the training process**:  Following your suggestion, we blended all the training data from the three stages (not the expanded ones in the above study) and trained the model in a single stage. The results are as follows: we observed that the three-stage training process outperforms the all-in-one stage training on RAG by a significant margin, while showing marginally better or worse performance on >100K and 32K tasks.
>
> | model | all-at-once  | ours |
> | -------- | ------- | ------- |
> | Short (ChatRAG)  | 48.55 |  52.5 |
> | Long (32K)  | 40.69 |  39.41 |
> | Ultra long (>100K)  | 35.07 |  35.59 |
>
> - **Training Efficiency**:  The sequence lengths of samples in the 1st and 2nd stages are short, i.e., less than 4K tokens, while the sequence lengths of samples in the 3rd stage are long, extending up to 128K tokens. If these samples are blended together, the sequence length of the training batch is determined by the longest sample. Therefore, stage-wise training effectively reduces the amortized sequence length during training, thus improving training efficiency.
>
> - In addition to the improvement of downstream tasks accuracy, the three-stage training process simplifies the experimental process. For example, at stage-2, we only need to optimize the best results on RAG tasks, without worrying about the long context tasks. At stage-3, optimizing long-context accuracy while maintaining RAG performance without degradation is easier than acquiring the best accuracy  of long-context and RAG performance simultaneously.
>
> ---
> ---
>
> > “Evaluating tradeoffs with other LLM capabilities with standard, general benchmarks ...”
>
> - Following your suggestion, we evaluated our updated ChatQA-2-8B model on MMLU, MT-Bench, HumanEval, and GSM8K. Our findings show that (i) our model outperforms Llama3.1-8B-Instruct on the math benchmark GSM8K (ii) it underperforms on the knowledge-intensive MMLU and the coding benchmark HumanEval. Since it has not been aligned by DPO / PPO process as Llama3.1-8B-Instruct, the MT-Bench score is lower. We could apply the RLHF process to improve its MT-Bench score in future.
>
>
>  |                    | ChatQA-2-8B | Llama3.1-8B-Instruct  |
> |--------------------|:-----------------:|:-----------:|
> |   Short (ChatRAG)  |       53.79       |    48.79    |
> |     Long (32K)     |       42.05       |    42.42    |
> | Ultra long (>100K) |       35.18       |    33.17    |
> |      HumanEval     |       66.46       |    70.73    |
> |        MMLU        |       65.73       |    67.59    |
> |      MT-Bench      |       8.09       |     8.42    |
> |   GSM8K   |       87.41       |    83.70    |

---

> > ### Author Response · Authors · 2024-11-27
> >
> > Thank you once again for the detailed comments and valuable suggestions. They are truly helpful in improving the quality of our paper. We hope our response addresses your concerns. We would be happy to discuss any further questions you may have.

---

> > > ### Comment · Reviewer_TcQ9 · 2024-11-27
> > >
> > > Dear authors,
> > >
> > > Thank you for your response. Here are a couple of follow up questions given the new information:
> > >
> > > > Increasing SFT samples (128K to 1.6M) in the first stage boosts RAG performance (sometimes)
> > >
> > > I strongly recommend adding the previous 128K version to Table 8 to provide insight for how adding more short-context instruction data affects the overall performance. Here is a truncated table of RAG performances with different context length assembled from information scattered in the paper:
> > >
> > >
> > > | | Llama3-ChatQA-2-8B (1.6M) | Llama3-ChatQA-2-8B (128K) | Llama3.1-8B-Instruct |
> > > | --------- | -------------| -------------| ------------- |
> > > | Short (ChatRAG) |	**53.79** |  52.50	| 48.79 |
> > > | Long (32K) |	42.05 | 39.41	| **42.42** |
> > > | Ultra long (>100K) |	35.18 | **35.59** |	33.17 |
> > >
> > > The result is actually quite interesting. The increased SFT data is for stage 1, which should have sequence length of less than 4K. It makes sense then for the 1.6M variant to perform worse on the Ultra long tasks (InfiniteBench) while performing significantly better on the short tasks (ChatRAG). What's surprising is the improvement on the Long (32K) benchmark. Can the authors comment on the result?
> > >
> > > > all-at-once vs 3-stage
> > >
> > > The ablation study is great. Obviously it would be even better if the authors could pinpoint the benefit of a multi-stage approach. Here are some follow-up questions/comments:
> > > * The statement on training efficiency is false, since the mixing of different length samples can be performed batch-wise, such that the samples from a batch all have similar sequence length.
> > > * Regarding the per-stage performance optimization, can the authors then comment on how the all-at-once version is optimized? Did you perform the same hyperparameter sweep as the 3-stage variant? What metric or task performance did you adopt to select the best model?
> > >
> > > > Performance on general benchmark
> > >
> > > Thank you for reporting all the performance truthfully on various common benchmarks. A good method does not need to excel in all aspects but the tradeoff should be made clear to users.

---

> > > > ### Author Response · Authors · 2024-11-28
> > > >
> > > > Dear Reviewer,
> > > >
> > > > We appreciate your quick response and the thoughtful follow-up questions you’ve raised. We will address them below.
> > > >
> > > > ---
> > > >
> > > > > “I strongly recommend adding the previous 128K version to Table 8 to provide insight for how adding more short-context instruction data affects the overall performance.”
> > > > - Many thanks for your suggestion. We will add the results in the final draft after completing the evaluation.
> > > > - Our previous trials suggest that the  benchmark results like HumanEval (coding) and GSM8K (math) depend on the amount of coding and math data in SFT, which is expected. The 128K version SFT data doesn’t have a good amount of coding and math-related data, so numbers could be low.  On the positive side, we obtained improved GSM8K/HumanEval by including abundant code and math data in the 1st stage of SFT, i.e.,  replacing the 1st stage 128K SFT data to 1.6M SFT data.  This improvement is orthogonal to the second and third stages of SFT, which focus on refining RAG and long-context capabilities.
> > > >
> > > > ---
> > > >
> > > > > “It makes sense then for the 1.6M variant to perform worse on the Ultra long tasks (InfiniteBench) while performing significantly better on the short tasks (ChatRAG). What's surprising is the improvement on the Long (32K) benchmark. Can the authors comment on the result?”
> > > > - We break down the average score on the Long (32K) benchmark across different tasks:
> > > >
> > > > |                           |  Avg. |   QMSum  |  Qasper |  QuALITY |  MuSiQue  |  HotpotQA  |  MultiFieldQA |
> > > > |---------------------------|:-----:|:-----:|:-----:|:-----:|:-----:|:-----:|:-----:|
> > > > | Llama3-ChatQA-2-8B (128K) | 39.41 | 11.64 | 28.85 | 63.35 | 27.81 | 53.81 | 51.02 |
> > > > | Llama3-ChatQA-2-8B (1.6M) | 42.05 | 13.58 | 29.56 | 67.65 | 32.99 | 58.18 | 50.33 |
> > > >
> > > > - The major improvements stem from QMSum, QuALITY, MuSiQue, and HotpotQA. Specifically, QMSum is a query-based summarization dataset, while QuALITY is a multiple-choice QA dataset. HotpotQA and MuSiQue are multi-hop QA datasets that require reasoning and answering questions based on multiple supporting documents. We think that the diversity of the 1.6M SFT data, in terms of task types, contributes to the improved performance across these  Long (32K) tasks.
> > > >
> > > > ---
> > > >
> > > > > “The statement on training efficiency is false, since the mixing of different length samples can be performed batch-wise, such that the samples from a batch all have similar sequence length.”
> > > > - Many thanks for the comment. We agree that mixing multiple datasets can be achieved batch-wise by bucketing samples of similar lengths from different datasets. However, implementing such a dataloader with a good mixing behavior requires much more engineering efforts, given the complexity of blending many datasets, each with varying average lengths (from 4K to >100K length), blending ratios, and sample counts. Simply sorting samples by lengths and loading batches with similar sequence lengths may result in poor mixing and "zig-zag" gradient updates, where batches predominantly consist of samples from one or only a few datasets.
> > > >
> > > > ---
> > > >
> > > > > “Regarding the per-stage performance optimization, can the authors then comment on how the all-at-once version is optimized? Did you perform the same hyperparameter sweep as the 3-stage variant? What metric or task performance did you adopt to select the best model?”
> > > >
> > > > - We observe that the primary performance optimization for instruction-tuning lies in the inclusion or exclusion of specific datasets in the training blend. In contrast, adjusting the blending weights has a much smaller impact on performance. For the all-at-once SFT approach, we use the same optimized list of datasets as in the 3-stage variant.
> > > >
> > > > - Given the experimental cycle and the limited time available during the rebuttal period, we tested multiple all-at-once versions using different blending ratios of Stage 1, 2, and 3 datasets. Here, we report the version that achieved the highest average score of Short (ChatRAG), Long (32K), and Ultra-Long (>100K).
> > > >
> > > > ---
> > > >
> > > > We hope our response addresses your questions. Many thanks again for your response.

---

> > > > ### Author Response · Authors · 2024-12-02
> > > >
> > > > Dear Reviewer,
> > > >
> > > > As the discussion period nears its end, we are wondering whether our recent response has addressed your follow-up questions. Please don't hesitate to let us know if you need any further clarification. We appreciate your kind consideration.
> > > >
> > > > Thank you very much,
> > > >
> > > > The Authors

---

### Official Review · Reviewer_teFx · 2024-11-04

**Soundness:** 3
**Presentation:** 3
**Contribution:** 2
**Rating:** 6
**Confidence:** 3

**Summary:**

This work develops ChatQA 2, a long context LLM based on llama3-8k model that matches the performance of the April version of OpenAI GPT4 models and outperform SOTA open-source models. This work introduces a two-stage adaptation pipeline including long context continued training and multi-stage instruction finetuning. The developed model is then evaluated on short-context RAG, medium context up to 32k and long context above 100k benchmarks to show the superior performances than the existing open-source models.

**Strengths:**

- The developed ChatQA2 model shows promising performances than existing open-source baselines and the April version of OpenAI GPT model.

- The evaluation from short to long contexts are comprehensive and convincing.

- This work presents a detailed data and training recipe that enables a good-performing long-context model, bridging the gap between the commercial sector and open-source community for long context modeling.

**Weaknesses:**

- The authors should clearly articulate the relationship of this work and ChatQA 1.5, right now it was described as both a baseline (as shown in Table 4) and confusingly as "the first version of this work", i.e. the same work. If ChatQA1.5 is indeed a baseline, then it should be listed in all the tables and result analysis.

- Some details of the computational experiment are missing in section 3: how many epochs and tokens were trained in each stage? How much does this impact the results? For example, there is a discussion to the potential reason of low longbench scores due to under-training. The authors should present the trend of higher scores with more training epochs (in supplementary information).  Please also provide the data, and training for reproducibility purposes.

- The authors should improve the comprehensiveness of their ablation studies and results analysis: (1) The authors should also evaluate the retrieval-based tasks in infinitebench to further confirm the NIAH superior performances as shown at the beginning. (2) The section 5.2 is a bit lacking: the RAG numbers seem not being discussed. (3) The authors should add an ablation study to support their claim that started at line #177, i.e. using an alternative document breaker helps the training.

- Minor comments: (1) the authors should add citations of scrolls and longbench when they are mentioned in line # 252 and line # 253. (2) There seems to be a typo in line #261: "as it is vasincluded" seems a bit confusing. (3) The authors should keep the bold texts consistent in all tables, right now some Llama-3.1-128k models are made bold but some are not. For example, I am not sure why 48.15 is made bold in Table 3, but not 49.92 (for 3.1-70b-instruct).

**Questions:**

1. At several places, e.g. line 144, 168, 222, in this manuscript, the authors mentioned "the first version of this work.", what does that mean? Please clarify.

2. What are the key new insights that this work offer, considering that the Llama3.1-128k model was published late July? I am not complaining about anything, but mainly want to make sure the technical contributions of this work. I will appreciate the answer from the authors.

3. Around line #188, the authors mentioned "where the model is initially trained on 128k high-quality instruction following datasets", while also mentioning " these two stages involve relatively short contexts", what does that mean, why 128k high-quality datasets are short contexts? Is there a typo?

4. In the long sequence SFT data, the authors used NarrativeQA for synthetic data generation. Are there potential data contamination in the benchmarks?


5. Minor questions: line # 263: why are the documents segmented into 300-word chunks? Does the size matter for the benchmark results?

**Details Of Ethics Concerns:**

N.A.

---

> ### Author Response · Authors · 2024-11-27
> **Official Comment by Authors (Part 1/2)**
>
> Many thanks for your comments and feedback. They are really helpful to improve the quality of our work. We will address your comments in the following:
>
> ---
>
> > “The authors should clearly articulate the relationship of this work and ChatQA 1.5, right now it was described as both a baseline (as shown in Table 4) and confusingly as "the first version of this work", i.e. the same work. If ChatQA1.5 is indeed a baseline, then it should be listed in all the tables and result analysis.”
> - We apologize for the confusion. "The first version of this work" does not refer to ChatQA 1.5 but rather to the initial version published at arXiv on July 19. We have updated the draft to avoid confusion.
> - ChatQA 1.5 is a short-context model that supports only up to 8K tokens. Therefore, it is included only in Table 4, as the other long-context tasks require models capable of handling 32K or >100K context windows.
>
> ---
>
> > “Please also provide the data, and training for reproducibility purposes.”
> - Thanks for bringing up this question. For reproducibility, we released model weights, evaluation data, and training data: https://chatqa2-project.github.io/ . We believe it is a useful contribution to the open LLM research.
>
> ---
>
> > "Some details of the computational experiment are missing in section 3: how many epochs and tokens were trained in each stage? How much does this impact the results? For example, there is a discussion to the potential reason of low longbench scores due to under-training. The authors should present the trend of higher scores with more training epochs (in supplementary information)."
> - This is a good question. During the continued pretraining stage, our model is trained on just 8B tokens extracted from a large corpus, with only one epoch of training. We intentionally limited the training data to 8B tokens, as this amount was sufficient for the model to pass the "Needle in a Haystack" test using "San Francisco" as the needle. Note that, Llama3.1-Instruct could pass the "passkey" test, but not the more challenging "San Francisco" one (see Figure 1 vs. Figure 3).
>
> - The three-stage SFT process involves blending multiple datasets, with each dataset potentially being trained for a different number of epochs, depending on its blending ratio and number of samples. For instance, the statistics for the datasets used in Stage 2 are provided in Appendix C of updated draft, which adds complexity to the analysis.
>
> - However, our findings indicate that increasing the amount of SFT data plays a very crucial role in improving performance. During the rebuttal phase, we trained another 8B ChatQA 2 model by **expanding the first-stage SFT dataset from 128K samples to 1.6M samples**. This resulted in an improvement in the score on long-context tasks (within 32K context window) from 39.41 to 42.05, bringing it very close to Llama3.1-8B-Instruct. Please see the table below for details. These results clearly demonstrate the importance of leveraging large volumes of SFT data.
>
> | model | Llama3-ChatQA-2-8B (128K samples)  | Llama3.1-8B-Instruct |  Llama3-ChatQA-2-8B (1.6M samples)  |
> | -------- | ------- | ------- | ------- |
> | scores  | 39.41   | 42.42  |  42.05  |
>
> ---
> ---
>
> > “The authors should improve the comprehensiveness of their ablation studies and results analysis: (1) The authors should also evaluate the retrieval-based tasks in infinitebench to further confirm the NIAH superior performances as shown at the beginning. (2) The section 5.2 is a bit lacking: the RAG numbers seem not being discussed. (3) The authors should add an ablation study to support their claim that started at line #177, i.e. using an alternative document breaker helps the training.”
>
> -  (1) We report the three retrieval based tasks in InfiniteBench below. Aligned with our superior NIAH scores, all our models get 100% accuracy for number_string and passkey tasks and our Llama3-ChatQA-2-70B get a better score than GPT-4-Turbo on kv_retrieval task.
> | model | kv_retrieval    | number_string |  passkey  |
> | -------- | ------- | ------- | ------- |
> | GPT-4-Turbo  | 89   | 100  |  100  |
> | Llama3-ChatQA-2-70B |  91  | 100   | 100    |
> | Llama3-ChatQA-2-8B    | 72    | 100 | 100 |
>
> - (2) Many thanks for the reminder. In Section 5.2, the RAG numbers use the default RAG setting (top-5 with chunk size 1200). The RAG shows it’s worse than the full context scores. However, in Table 5, we show that RAG numbers can be better than full long context solutions if we increase the number of chunks in the prompt. We add discussion and also reference Section 6 in the update draft.
>
> - (3) Due the limited time of rebuttal, we trained our model on 2B tokens with different separators and the NIAH shows that using document breaker “<s>” is much better. The figure can be found below. We’ve added this ablation in Appendix B.
> https://docs.google.com/document/d/1UaoaVDlyj9jRKXtFPls_bUErHAVa50_FPisYXWW6aWE/edit?usp=sharing

---

> ### Author Response · Authors · 2024-11-27
> **Official Comment by Authors (Part 2/2)**
>
> This is a follow-up response to the remaining questions.
>
> > “Minor comments: (1) the authors should add citations of scrolls and longbench when they are mentioned in line # 252 and line # 253. (2) There seems to be a typo in line #261: "as it is vasincluded" seems a bit confusing. (3) The authors should keep the bold texts consistent in all tables, right now some Llama-3.1-128k models are made bold but some are not. For example, I am not sure why 48.15 is made bold in Table 3, but not 49.92 (for 3.1-70b-instruct).”
> - (1) Thanks for the reminder. We add them in the updated draft.
> - (2) and (3). Thanks for pointing them out. We have fixed them in the draft.
>
> ---
> ---
>
> Questions:
>
> > “What are the key new insights that this work offer, considering that the Llama3.1-128k model was published late July? I am not complaining about anything, but mainly want to make sure the technical contributions of this work. I will appreciate the answer from the authors.”
>
> - Thank you for bringing up this question. Our work was released on July 19, prior to the release of Llama3.1-128k. While both can be considered concurrent efforts, ChatQA 2 offers unique technical contributions in the following ways:
>
> - **Continued pre-training**:  Llama3.1-128k was trained on a proprietary dataset of 800B tokens during a continued pretraining stage, which increased the supported context window from 8K to 128K. In contrast, our ChatQA2-128k was trained on a publicly available dataset of 8B tokens during continued pretraining, which is 100 times smaller than Llama3.1-128k. The sample efficiency of our training process comes from (i) upsampling long-context documents in the pretraining corpus and (ii) packing different documents using special characters, such as “<s>,” rather than the reserved beginning and ending tokens <BOS> and <EOS> in Llama3. Both methods enable the model to quickly adapt to longer-context inputs.
>
> - **Instruction-tuning stage**:  Our goal is to provide an open-source recipe by sharing the full pipeline and training data, enabling others to customize and build upon it.  Specifically, we curate the SFT datasets, especially long context SFT data, and make them publicly available to support open LLM research. In contrast, for Llama3.1-128K, we found no publicly available information regarding the composition of its SFT datasets, preventing a direct comparison of methodology and data curation approaches. Empirically, ChatQA 2 demonstrates comparable performance across various tasks with different context lengths.
>
> ---
>
> > “Around line #188, the authors mentioned "where the model is initially trained on 128k high-quality instruction following datasets", while also mentioning " these two stages involve relatively short contexts", what does that mean, why 128k high-quality datasets are short contexts? Is there a typo?”
> -  Sorry for the confusion. This “128k high-quality instruction following dataset” here refers to the total number of samples in the blended dataset, rather than the sequence length of the sample. We clarify this in the updated draft.
>
> ---
>
> > “In the long sequence SFT data, the authors used NarrativeQA for synthetic data generation. Are there potential data contamination in the benchmarks?”
> -  Thank you for your question. Since we used NarrativeQA for synthetic data generation, we intentionally excluded it from the evaluation benchmarks in this submission to avoid any potential data contamination. We clarify this point in the updated manuscript.
>
> ---
>
> > “Minor questions: line # 263: why are the documents segmented into 300-word chunks? Does the size matter for the benchmark results?”
> - Thank you for your question. We initially selected 300-word chunks because this size is widely supported by text-embedding models, particularly BERT-based ones. For embedding models capable of supporting longer contexts (e.g., E5-mistral-7B), we conducted an ablation study to evaluate the impact of chunk size. In Figure 2, we present the results of this study, which explored different chunk sizes {300, 600, 1200} alongside varying top-k values {5, 10, 20, 40}. When the total number of tokens is fixed at 6K, we observed that a chunk size of 600 performs the best, with 300-word chunks slightly lagging behind. However, as demonstrated in the figure, the chunk size has considerably less impact compared to the total number of tokens included in the prompt.
>
> ---
> ---
>
> Many thanks again for your detailed comments and valuable suggestions. They are truly helpful in improving the quality of our paper. We hope our response addresses your major concerns. We would be happy to discuss any further questions you may have.

---

> > ### Author Response · Authors · 2024-11-30
> >
> > Dear Reviewer,
> >
> > Thank you once again for your detailed comments and suggestions. We have incorporated them into the draft accordingly and look forward to hearing whether our responses address your concerns.
> >
> > Thanks,
> >
> > Authors

---

> ### Author Response · Authors · 2024-12-02
>
> Dear Reviewer,
>
> As the discussion period comes to a close, we would like to take this opportunity to express our gratitude for your detailed comments and specific questions. We have tried our best to address your questions in an adequate manner. Please let us know if you require any further clarification.
>
> Many thanks,
>
> The Authors

---

> ### Comment · Reviewer_teFx · 2024-12-03
>
> I appreciate authors' explanation and clarification of the manuscript, as well as sharing the publicly released data and model weights. This improve the clarity of the work, and therefore I am happy to raise the score to 6.
>
> My only doubt now is that according to the data repo and website, it says https://huggingface.co/nvidia/Llama3-ChatQA-2-70B , which I am not sure if it's aligned with our double-blind review policy. Our area chairs or program chairs may want to confirm on that. Thank you.

---

> ### Author Response · Authors · 2024-12-03
>
> Dear Reviewer,
>
> Many thanks for kindly raising the score. We provided the website link as you requested: "Please also provide the data, and training for reproducibility purposes."  Both the model weights and data are substantial in size, so we didn't host them elsewhere during the short rebuttal period. We sincerely appreciate your understanding.
>
> Many thanks,
>
> The authors

---

### Official Review · Reviewer_UtmQ · 2024-11-04

**Soundness:** 2
**Presentation:** 3
**Contribution:** 3
**Rating:** 6
**Confidence:** 3

**Summary:**

This paper introduces ChatQA 2, an open-source long-context language model with a 128K token context window, aiming to achieve performance comparable to GPT-4-Turbo. By leveraging a continue pretraining and a three-stage instruction tuning process, ChatQA 2 demonstrates improved accuracy and robustness in handling large volumes of context and consistently outperforms direct long-context models when using RAG with sufficient top-k chunks.

**Strengths:**

1. This paper introduces a significant enhancement to Llama3-70B’s long-context processing abilities, extending its context window from 8K to 128K tokens. Compared to Llama3.1with 128K context window, the proposed ChatQA2 achieves better performance.

2. This paper proposes a three-stage instruction tuning, and collect a long SFT dataset to enhance the model's capability to handle very long context sequences up to 128k tokens.

3. The authors propose utilizing the latest long-context retriever to address challenges in the current RAG pipeline for LLMs, specifically the limitations of top-k chunk-wise retrieval.

**Weaknesses:**

1. Compared to Llama3.1, ChatQA 2 shows improved performance on InfiniteBench, which includes real-world long-context understanding tasks beyond a 100K context window. However, on long-context benchmarks within 32K tokens, ChatQA 2 does not outperform Llama3.1, possibly due to its instruction tuning being focused on long SFT dataset. The author should also pay attention to the performance of 32K tokens.

2. The long-context open-source model ChatQA 2 presented in this paper achieves strong performance, though it lacks some innovation. The synthetic dataset used in the long SFT dataset could be described in more detail.

**Questions:**

1. Will the collected long SFT dataset utilized in the three-stage instruction tuning be open source?

2. Will the ChatQA 2 model be open source?

---

> ### Author Response · Authors · 2024-11-26
>
> Many thanks for your detailed comments and feedback. We will address your comments in the following:
>
> ---
>
> > "Compared to Llama3.1, ChatQA 2 shows improved performance on InfiniteBench, which includes real-world long-context understanding tasks beyond a 100K context window. However, on long-context benchmarks within 32K tokens, ChatQA 2 does not outperform Llama3.1, possibly due to its instruction tuning being focused on long SFT dataset. The author should also pay attention to the performance of 32K tokens."
>
> -  Thank you for your insightful comment. We hypothesized that the primary reason ChatQA 2 underperforms compared to Llama 3.1 on long-context benchmarks within 32K tokens is the limited availability of high-quality open-source SFT datasets, especially when compared to the proprietary SFT datasets used in Llama 3.1 and Qwen2.
>
> - Our findings indicate that increasing the amount of SFT data plays a crucial role in improving performance. During the rebuttal phase, we trained another 8B ChatQA 2 model by expanding the first-stage SFT dataset from 128K samples to 1.6M samples. This resulted in an improvement in the score on long-context tasks (within 32K tokens) from 39.41 to 42.05, bringing it very close to Llama3.1-8B-Instruct. Please see the table below for details. These results clearly demonstrate the importance of leveraging large volumes of SFT data.
>
> | model | ChatQA-2-8B (128K SFT samples)   | Llama3.1-8B-Instruct |  ChatQA-2-8B (1.6M SFT samples)  |
> | -------- | ------- | ------- | ------- |
> | scores  | 39.41   | 42.42  |  42.05  |
>
> - Additionally, we want to emphasize that we used only 8B tokens for further pretraining the Llama 3.0 checkpoint to extend its context window to 128K, in contrast to the 800B tokens utilized for continued pretraining in Llama 3.1 for the same purpose.
>
> ---
>
> > "The long-context open-source model ChatQA 2 presented in this paper achieves strong performance, though it lacks some innovation. The synthetic dataset used in the long SFT dataset could be described in more detail."
> - Thank you for your constructive comment. In our study, we found that achieving state-of-the-art performance heavily depends on the curation of data. We believe that curating and sharing training data can provide benefits to the community comparable to those achieved by proposing innovative methods.
> - Yes, the NarrativeQA dataset includes summary paragraphs, questions, answers, and source long web pages. The summaries are human-generated based on the source web pages, while the question-answer pairs are annotated by humans using the summaries. To extend the context length, we inserted a summary into the corresponding long web page document at a random location, ensuring that sentence structure was not disrupted. This approach maintains the grounding of the question-answer pairs within the augmented long documents. We will include a detailed description in Section 3.2 in the updated draft.
> - Inspired by your suggestion, we could leverage the short context to retrieve relevant long web pages and apply a similar strategy to extend short context datasets to long contexts (e.g., 32K tokens) in future work.
>
> ---
>
> > “Will the collected long SFT dataset utilized in the three-stage instruction tuning be open source?” “Will the ChatQA 2 model be open source?”
> - Yes, we have open-sourced the model weights (both ChatQA-2-8B and 70B), training data, and evaluation data at:  https://chatqa2-project.github.io/. We believe it is a useful contribution to the open LLM research community.
>
> Thank you once again for your valuable comments and helpful suggestions. We hope our response helps to address your concerns. If you have any additional questions, please don’t hesitate to let us know—we’d be happy to discuss them further.

---

> > ### Author Response · Authors · 2024-12-02
> >
> > Dear Reviewer,
> >
> > Many thanks again for your detailed comments and suggestions. We have incorporated them into the updated version of our submission and hope that our response addresses your questions. As the discussion period comes to a close, please don’t hesitate to let us know if you have any remaining questions or concerns. We look forward to hearing from you.
> >
> > Best,
> >
> > Authors

---

### Official Review · Reviewer_BLGw · 2024-11-07

**Soundness:** 1
**Presentation:** 1
**Contribution:** 2
**Rating:** 5
**Confidence:** 4

**Summary:**

This paper presents a way to train a short context base model to handle long context. In involves positional encoding (RoPE) to allow long context handling, pretraining on long context data and RAG. The trained performance on llama 70b shows comparable results to commercial LLMs in long context.

**Strengths:**

The performance looks good.

**Weaknesses:**

- The method is not clearly presented. I don't understand what fundamental changes of RAG or finetuning that makes the long context performance better
- The ablation study of the training process / design is missing. I don't understand the takeaway message from this paper

**Questions:**

- More details in the method. The 3.1 and 3.2 looks quite common. Since the authors emphasizes RAG, what specific changes have the authors made? And how RAG is used in the training?
- I am a bit confused by the highlight of this paper. I suppose the authors want to present a training pipeline for long context. But I don't quite understand the ablation of why certain choices are made.

---

> ### Author Response · Authors · 2024-11-26
> **Official Comment by Authors (Part 1/2)**
>
> Thank you for your comment. We would like to start by clarifying the goal and takeaways of this work, as requested in your review. After that, we will address your specific comments in detail.
>
> **Goal**:  In previous studies, retrieval-augmented generation (RAG) and long context windows are two rival methods that enable LLMs to handle long inputs exceeding the regular context window (e.g., 8K tokens). Each method has its own pros and cons — for example, long context models outperform RAG for text summarization tasks, while RAG reduces inference costs by lowering the total number of tokens fed into the LLM. Therefore, it is very useful to train a single performant LLM with a long context window (e.g., 128K) that also excels at RAG to address diverse downstream task requirements. Then, users can choose to feed either the whole long document or top-k retrieved chunks to the LLM, depending on the type of questions (e.g., information-seeking QA vs. summarization), the length of the documents, and the computational budget at inference. In this work, we build ChatQA 2 that can match leading proprietary and open-weights models on various long-context and RAG tasks, while sharing reproduction recipe and releasing the training data at: https://chatqa2-project.github.io/ (unprecedented in the literature).
>
> **Takeaway**: Building on the long-context model ChatQA2-128K, we explore how to effectively combine RAG and long-context methods to leverage the strengths of both approaches. For instance, Figure 2 illustrates an intriguing trade-off between downstream task accuracy and inference cost (measured by the total tokens in the context window) when applying RAG with the proposed ChatQA2-128K. Notably, as the volume of top-k retrieved context increases (total tokens > 12K), RAG's accuracy not only matches but even surpasses that of the full long-context solution, while still requiring fewer input tokens within the LLM's context window (e.g., 12K vs. 32K).
>
> ---
>
> We will address your specific comments in detail in the following.
>
> > “The method is not clearly presented. I don't understand what fundamental changes of RAG or finetuning that makes the long context performance better.” “More details in the method. The 3.1 and 3.2 looks quite common. Since the authors emphasizes RAG, what specific changes have the authors made? And how RAG is used in the training?”
>
> - Continued pretraining: Section 3.1 describe continued pretraining for context window extension. Compared to the literature, Llama3.1-128k (released after our work) was trained on a proprietary dataset of 800B tokens during a continued pretraining stage. In contrast, our ChatQA2-128k was only trained on 8B tokens extracted from publicly available dataset of (i.e., Slimpajama) during continued pretraining, which is 100 times smaller than Llama3.1-128k. The data efficiency of our training process comes from (i) upsampling long-context documents in the pretraining corpus (Fu et al. (2024)), and (ii) packing different documents using special characters, such as “<s>,” rather than the reserved beginning and ending tokens <BOS> and <EOS> in Llama3. Both methods enable the model to quickly adapt to longer-context inputs. Also note that previous work (Fu et al. 2024) built on open-sourced datasets haven’t demonstrated state-of-the-art performance on real-world downstream tasks.
>
> - Instruction-tuning: The **fundamental changes** lie in our  three-stage instruction-tuning algorithms in section 3.2. In stage-1, we first apply general SFT to enhance the general capabilities of LLM (e.g.,  instruction-following). In stage-2, we blend general SFT data from stage-1 with the additional datasets that can significantly enhance RAG performance, e.g., QA and conversational QA datasets with retrieved or provided passage / context from long documents or large corpus (e.g., Wikipedia).  In stage-3, we blend all training data from stage 1 & 2 (to avoid forgetting) with additional long-context SFT datasets that can enhance long-context performance. The three-stage training pipeline is designed to maintain the capabilities obtained from previous stages while acquiring the new capability. We will include an ablation study in the comment (Part 2/2).  Notably, we curate and open-source the instruction-tuning datasets for long context tasks, offering a fully reproducible recipe for achieving state-of-the-art results on ultra-long context tasks exceeding 100K tokens and RAG tasks.

---

> ### Author Response · Authors · 2024-11-26
> **Official Comment by Authors (Part 2/2)**
>
> This is a follow-up response to the remaining questions.
>
> >  “The ablation study of the training process / design is missing..”
> - **Ablation of the training process**:  Following your suggestion, we blended all the instruction-tuning data from the three stages and trained the Llama-3.0-8B base in a single stage. The results are as follows: we observed that the three-stage training process outperforms the all-in-one stage training on RAG by a significant margin, while showing marginally better or worse performance on >100K and 32K tasks.
>
> | model | all-in-one stage  | three-stage |
> | -------- | ------- | ------- |
> | Short (ChatRAG)  | 48.55 |  52.5 |
> | Long (32K)  | 40.69 |  39.41 |
> | Ultra long (>100K)  | 35.07 |  35.59 |
>
> - **Training Efficiency**:  The sequence lengths of samples in the 1st and 2nd stages are short, i.e., less than 4K tokens, while the sequence lengths of samples in the 3rd stage are long, extending up to 128K tokens. If these samples are blended together, the sequence length of the training batch is determined by the longest sample. Therefore, stage-wise training effectively reduces the amortized sequence length during training, thus improving training efficiency.
>
> - In addition to the improvement of downstream tasks accuracy, the three-stage training process simplifies the experimental process. For example, at stage-2, we only need to optimize the best results on RAG tasks, without worrying about the long context tasks. At stage-3, optimizing long-context accuracy while maintaining RAG performance without degradation is easier than acquiring the best accuracy  of long-context and RAG performance simultaneously.
>
> ---
>
> Thank you once again for your review. We appreciate your feedback on our work and will update the draft accordingly. We hope our response can help address your major concerns. Please let us know if you have any further questions. We would be happy to discuss them further with you.

---

> ### Author Response · Authors · 2024-11-30
>
> Dear Reviewer,
>
> Thank you once again for your comments and suggestions. We have incorporated them into the draft accordingly. We look forward to hearing if our responses address your concerns and would be happy to discuss further if you have any additional questions.
>
> Best,
>
> Authors

---

> ### Author Response · Authors · 2024-12-02
>
> Dear Reviewer,
>
> Thank you so much for your reply and kindly update the score.
>
> > "My current concern is the contribution significance of this method. It makes sense that mitigating the test distribution shift by continued training on distributionally similar corpus could help. But this is proven in too many cases such that the takeaway message from this work is limited."
>
> The contribution of this method is not limited to mitigating test distribution shifts through continued training on a distributionally similar corpus.
>
> We will elaborate on this in the following discussion.
>
> 1.  Addressing long-context test tasks requires training on a long-context corpus and long-context SFT data. However, the primary challenge lies in preserving the performance of other tasks (e.g., short-context RAG) while enhancing the model's long context capabilities. This essential requirement underscores the rationale behind the proposed three-stage instruction tuning approach. For instance, the third stage, long-context SFT, is designed to enhance the model's long-context capabilities, while preserving the RAG capability acquired during the second stage. As a result, our ChatQA-2-70B-128K successfully achieves SOTA performance on both long context tasks and RAG tasks.
>
>
> 2. Importantly, this success empowers a new capability of the model without explicit training. Note that, (i) the model's RAG-related training data are all short-context (<4K tokens). (ii) The long-context training data focus on book or long-document understanding without top-k retrieval. Despite the disjoint training data for two capabilities, the learned long-context capability effectively transfers to the RAG scenario. Specifically, the generation accuracy of our Llama3-ChatQA-2-70B consistently improves as the total number of retrieved tokens in the input increases from 3K to 24K tokens (using top-20 chunks, each with a chunk size of 1,200, as shown in Figure 2).  In contrast, previous RAG models are limited to short-context inputs (typically <4K tokens) due to the well-known precision-recall trade-off: a smaller top-k reduces the recall of relevant context, while a larger k improves recall but introduces excessive irrelevant context, potentially degrading generation accuracy. For examples, see Figure 1 in [1] Yu et al. (2024) and Table 5 in Xu et al. (2024).
>
> This indicates our model's ability to generalize effectively without requiring an explicit match between the training and test distributions. We hope this addresses your current concern, and we sincerely appreciate your kind consideration.
>
> [1] Yu et al. RankRAG: Unifying context ranking with retrieval-augmented generation in LLMs. NeurIPS 2024.
>
> [2] Xu et al. Retrieval meets Long Context Large Language Models. ICLR 2024.

---

### Meta-Review · Area_Chair_1SWC · 2024-12-11

**Metareview:**

(a) Scientific Claims and Findings
This paper introduces ChatQA 2, an open-source long-context language model with a 128K token context window, demonstrating improved accuracy and robustness in handling large volumes of context and consistently outperforming direct long-context models when using RAG with sufficient top-k chunks.

(b) Strengths of the Paper
The paper introduces a significant enhancement to Llama3-70B's long-context processing abilities, extending its context window from 8K to 128K tokens.
The paper proposes a three-stage instruction tuning and collects a long SFT dataset to enhance the model's capability to handle very long context sequences up to 128k tokens.
The authors propose utilizing the latest long-context retriever to address challenges in the current RAG pipeline for LLMs, specifically the limitations of top-k chunk-wise retrieval.

(c) Weaknesses of the Paper
Compared to Llama3.1, ChatQA 2 shows improved performance on InfiniteBench, which includes real-world long-context understanding tasks beyond a 100K context window.
The long-context open-source model ChatQA 2 presented in this paper achieves strong performance, though it lacks some innovation.

(d) Reasons for Accepting the Paper
The paper presents a detailed data and training recipe that enables a good-performing long-context model, bridging the gap between the commercial sector and the open-source community for long context modeling.
The developed ChatQA2 model shows promising performances than existing open-source baselines and the April version of the OpenAI GPT model.
The evaluation from short to long contexts is comprehensive and convincing.

**Additional Comments On Reviewer Discussion:**

Rebuttal Period Discussion and Changes
During the rebuttal period, the authors clarified the goal and takeaways of their work, emphasizing the importance of the three-stage instruction-tuning process.  They also provided more details about the computational experiment, including the number of epochs and tokens trained in each stage.  The authors further clarified the relationship between ChatQA 2 and ChatQA 1.5 and defended the significance of their contributions.

Points Raised by Reviewers and Author Responses
Reviewer BLGw questioned the significance of the method and its contribution beyond mitigating test distribution shifts.  The authors responded by emphasizing the importance of preserving performance on other tasks while enhancing long-context capabilities, a key rationale behind their three-stage instruction tuning approach.
Reviewer UtmQ pointed out the lack of innovation in the long-context open-source model ChatQA 2 and requested more details about the synthetic dataset used in the long SFT dataset.  The authors responded by emphasizing the importance of data curation and sharing training data, which they believe can provide significant benefits to the community. They also explained how they extended the context length in the NarrativeQA dataset.
Reviewer teFx asked for clarification on the relationship between ChatQA 2 and ChatQA 1.5 and requested more details about the computational experiment.  The authors clarified that "the first version of this work" refers to the initial version published at arXiv on July 19, not ChatQA 1.5. They also provided more details about the computational experiment, including the number of epochs and tokens trained in each stage.
Weighing in Each Point for the Final Decision
The reviewers' concerns about the novelty and the significance of the contributions of the paper are valid, and the authors' response did not fully alleviate the concerns.
The authors' clarification on the relationship between ChatQA 2 and ChatQA 1.5 and their detailed explanation of the computational experiment are helpful, but they do not significantly change the assessment of the paper's contributions.
The authors' response to the question about the lack of innovation in ChatQA 2 is reasonable, but it does not fully address the reviewer's concern.
Overall, the strengths of the paper, such as the detailed data and training recipe, the promising performance of the ChatQA2 model, and the comprehensive evaluation, outweigh the reviewers' concerns about the novelty and significance of the paper's contributions, leading to a decision of acceptance.

---

### Decision · Program_Chairs · 2025-01-22

Accept (Poster)